# An Educational Study Promoting the Delivery of Transcranial Doppler Ultrasound Screening in Paediatric Sickle Cell Disease: A European Multi-Centre Perspective

**DOI:** 10.3390/jcm9010044

**Published:** 2019-12-24

**Authors:** Baba P. D. Inusa, Laura Sainati, Corrina MacMahon, Raffaella Colombatti, Maddalena Casale, Silverio Perrotta, Paola Rampazzo, Claire Hemmaway, Soundrie T. Padayachee

**Affiliations:** 1Paediatric Haematology Department, Evelina London Children’s Hospital, Guy’s & St Thomas’ NHS Foundation Trust, St Thomas Street, London SE1 7EH, UK; 2Clinic of Pediatric Hematology-Oncology, Department of Child and Maternal Health, Azienda Ospedaliera-Università di Padova, Via 8 Febbraio 1848, 2, 35122 Padova, Italy; laura.sainati@unipd.it (L.S.); rcolombatti@gmail.com (R.C.); 3Paediatric Haematology, Our Lady’s Children’s Hospital, Cooley Rd, Crumlin, D12 N512 Dublin, Ireland; corrina.mcmahon@olchc.ie; 4Università degli Studi della Campania, Luigi Vanvitelli, Via Luciano Armanni, 14-20, 80138 Napoli, Italy; casale.maddalena@gmail.com (M.C.); silverio.perrotta@unicampania.it (S.P.); 5Department of Neurosciences, Azienda Ospedaliera-Università di Padova, Via 8 Febbraio 1848, 2, 35122 Padova, Italy; patrizia.rampazzo@unipd.it; 6Department of Haematology, Queen’s Hospital, Rom Valley Way, Romford RM7 0AG, UK; c.hemmaway@nhs.net; 7Ultrasonic Angiology Department, Guy’s & St Thomas’ NHS Foundation Trust, London SE1 9RT, UK; Soundrie.Padayachee@gstt.nhs.uk

**Keywords:** sickle cell disease, transcranial Doppler (TCD), screening, education, stroke

## Abstract

Background: Effective stroke prevention in sickle cell disease (SCD) is recommended for children with sickle cell anaemia. Effective implementation relies on the correct stratification of stroke risk using Transcranial Doppler Ultrasound (TCD), prior to committing children to long-term treatment with transfusion. Nevertheless, less than 50% of children with SCD in Europe receive annual TCD—one of the reasons being a lack of trained personnel. The present European multi-centre study was designed to promote the standardisation and delivery of effective screening. Methods: Fifty-five practitioners from differing professional backgrounds were recruited to the TCD training program. The impact of the training programme was evaluated in three European haematology clinics by comparing stroke risk classification and middle cerebral artery time-averaged maximum velocity (TAMMV) obtained from a cohort of 555 patients, before and after training. Results: 42% (23/55) of trainees successfully completed the program. The TAMMV, used to predict stroke risk at each Centre, demonstrated the highest values in Centre 3 (*p* < 0.0001) before training. The imaging-TCD TAMMV was also higher in Centre 3 (*p* < 0.001). Following training, the TAMMV showed closer agreement between centres for both imaging-TCD and non-imaging TCD. The stroke risk distribution of children at each centre varied significantly before training (*p* < 0.001), but improved after training (Fisher’s Exact: no treatment = 5.6, *p* = 0.41, treatment = 13.8, *p* < 0.01). The same consistency in stroke risk distribution following training was demonstrated with both non-imaging and imaging-TCD data. Conclusion: The attainment of competency in stroke screening using transcranial Doppler scanning (TCD) in sickle cell disease is more feasible for professionals with an ultrasound imaging background. A quality assurance (QA) system is required to ensure that standards are maintained. Further work is in progress to develop an achievable and reproducible QA program.

## 1. Introduction

Recent studies on the impact of migration on the geographical distribution of the HbS allele have highlighted sickle cell disease (SCD) as a global public health issue [1,2,3,4]. Although considered a “rare disease” due to its global frequency, in the 28 countries of the European Union, SCD is the most common single gene genetic disease in France and the United Kingdom and its frequency is steadily rising in many other countries of central and southern Europe [5,6,7,8]. Establishing clinical services for rare disorders is encouraged, and the importance of identifying areas with poor access to health services is well-recognised [9]. Several countries have organised a specific response in order to meet the needs of the growing number of patients with SCD, including establishing newborn screening and stroke risk screening programmes [7,8,10]

A worldwide SCD stroke risk screening programme was recommended following the Stroke Prevention Trial in Sickle Cell Anemia (STOP) which provided Level 1 evidence of the reduction in primary stroke risk following blood transfusion for children with a high risk of stroke [11]. Stroke risk was based on the presence and severity of intracranial stenosis which, was reliably estimated using Transcranial Doppler Ultrasound (TCD). The TCD technique involves the meticulous interrogation of the basal cerebral arteries, with frequent optimisation of the ultrasound transducer orientation to ensure measurement of the highest blood velocity, which is related to stroke risk [11]. Therefore, TCD screening is recommended as a standard of care for children with SCD starting at an age of two years [10,12,13]. A substantial variation in the access to TCD screening has been reported, varying from as low as <50% [14,15], to as high as >90% in children with SCD [16,17]. There are a number of reasons why access to TCD services remains a challenge; funding is a major issue, but even with adequate funds, issues remain due to a lack of skills and knowledge, particularly with respect to adequately trained TCD operators. The absence of a formal training centre outside of the USA contributes to the failure to implement the recommendation. A complicating factor is that TCD screening can be delivered using two types of equipment described as non-imaging or imaging TCD [18]. Most centres use the non-imaging approach, as described in the STOP trials, which is a “blind technique” where there is no guiding anatomical information and thus relies heavily on operator experience. Some centres now use imaging TCD, which provides anatomical information enabling the Circle of Willis to be visualised and so facilitates the identification of the basal cerebral arteries and orientation of the Doppler beam when acquiring blood velocities. There have, however, been reported discrepancies between the two techniques—resulting in concerns regarding the validity of using the same velocity thresholds for STOP classification [18,19,20,21,22,23]. 

The management of patients with SCD needs to be standardised [24]. Nevertheless, it must be tailored according to the characteristics of the affected population in each country and the availability of healthcare resources, so that public health issues such as service delivery and access to care can be areas of common policy for European countries. This prospective, multi-centre study focused on the provision and development of *TCD*. The primary study objective was to determine the effectiveness of the modular training programme in achieving the high level of scanning competency described in the STOP trial, irrespective of practitioner background and when using either non-imaging or imaging TCD. If successful, this modular educational program would facilitate the widespread availability of TCD screening and thus increase the number of children with SCD being screened with TCD. The secondary objectives were to identify technical factors that influenced the quality, and the standardisation of TCD assessment.

## 2. Methods

The study was registered with the Research Ethics Committee for Wales (ref 09/MRE09/38) and approval was obtained from the local Ethics Committees at each of the participating Hospitals. Written informed consent was obtained from the parents, and assent from the children participating in the study.

### 2.1. Modular TCD Training Programme

The modular TCD training programme was developed at the training centre in London and delivered to trainees at all three centres, comprising of a 2-day instructional course. Day 1 covered sickle cell disease: the clinical problem, relevant cerebral anatomy and haemodynamics, Transcranial Doppler theory and instrumentation, the principles of non-imaging, and imaging TCD scanning protocols. Scan protocols were demonstrated, followed by an afternoon of hands-on sessions scanning volunteers and patients. Day 2 covered the TCD surveillance programme, screening children with SCD and further hands-on sessions, finishing with a competency evaluation to assess the trainees’ practical and TCD data interpretation skill levels. This was followed by trainees scanning at their own hospital until they had collected a logbook of at least 40 scans (within a one year period), after which a scan review and repeat competency evaluation were performed.

### 2.2. Pre-Training TCD Practice at Each Centre

Prior to the training programme the practicing TCD scanning protocol and STOP classification were documented for each centre: Centre 1 (training Centre) performed both imaging and non-imaging TCD. Non-imaging TCD was performed according to the STOP protocol using STOP velocity thresholds for stroke risk evaluation (normal < 170 cm/s, conditional 170–199 cm/s and abnormal ≥ 200 cm/s). Imaging TCD was performed based on the STOP protocol, which included optimisation of transducer orientation according to the non-imaging protocol, but STOP velocity thresholds were reduced by 10% (normal < 155 cm/s, conditional 155–180 cm/s and abnormal ≥ 180 cm/s) based on guidance from the literature [11,12].Centre 2 used non-imaging TCD, only performed according to the standard STOP protocol.Centre 3 performed both imaging and non-imaging TCD. Non-imaging TCD was performed according to the STOP protocol and imaging TCD was performed according the conventional duplex scanning approach, which included Doppler angle correction, to adjust for any vessel misalignment with respect to the Doppler beam, in addition STOP velocity thresholds were also reduced by 10%.

### 2.3. Training

Staff performing TCD at the 3 centres and other professionals requiring training were entered on to the training course. Participation was voluntary, but all trainees were either actively involved in the delivery of TCD scanning or required training to enable them to do so at their local hospital. Trainees were from a range of professional backgrounds including clinical scientists, radiographers, paediatric haematologists, paediatric neurologists, surgeons and nurses—all of whom were involved in the care of children with SCD. The diversity of trainee professional backgrounds demonstrates the fact that service providers did not have a systematic approach to selecting the staff for training to undertake TCD scanning across the three centres, there was no requirement to the nature of trainees’ backgrounds. All TCD practitioners completed the modular training programme, which was designed to:Teach the fundamentals of TCD using imaging and non-imaging modalities.Instruct trainees on how to optimise imaging and non-imaging TCD assessment of the basal cerebral arteries to obtain the highest possible blood velocity (TAMMV).Instruct trainees in the use of STOP categories for stratifying stroke risk.

Clinical skills training was delivered on a one-to-one basis by experienced clinical scientists, performing scans on volunteer subjects and children with a range of TCD STOP classifications. Trainees had a range of previous TCD experience ranging from none to competent, those trainees with previous experience progressed through the training programme at a much faster rate than inexperienced professionals. Repeat scans by the clinical skills trainers provided quality assurance of the trainees’ scanning progress and accuracy.

### 2.4. Post-Training Standardised TCD Practice

A standard STOP protocol was used to obtain blood velocities from the middle, anterior and posterior cerebral and terminal internal carotid arteries using non-imaging TCD. The transducer orientation was optimised at every 2mm interval in each vessel to ensure that the highest, audible Doppler frequency signal was obtained. The same scanning approach and STOP classification was adopted for imaging TCD based on previous studies that have shown good agreement between velocity measured by imaging and non-imaging TCD modalities [25]. The colour flow anatomical information provided by imaging TCD was used to aid vessel localisation, after which the highest audible Doppler frequency was recorded following appropriate transducer optimisation. STOP thresholds were those described in the original STOP trial (normal <170 cm/s, conditional 170–199 cm/s, and abnormal >200 cm/s TAMMV).

TCD competency evaluations were performed at the end of the instructional course and 6–12 months later when a logbook of at least 40 TCD scans was completed. The logbook included ultrasound images of all patients scanned, stored electronically or on hard-copy. TCD practitioners were approved as competent when they demonstrated correct scanning technique, identification of the major basal cerebral arteries, correct optimisation to obtain the highest achievable velocity, correct velocity data interpretation and applying the correct STOP classification and surveillance interval. Competent trainees were also required to identify non-diagnostic scans, defined as when velocities could not be obtained due to either patient movement or signal attenuation.

### 2.5. Comparative Analysis of TCD Data Obtained before and after Training

The impact of the training programme was assessed by comparing STOP classifications and TCD velocity measurements obtained before and after TCD training in the three paediatric haematology clinics in London (England, UK; training centre), Crumlin (Ireland) and Padova (Italy). The data comprised of TCD scans performed up to one year before the TCD training programme was delivered and scans in the same cohort of patients performed after competency had been achieved. The pre-training TCD data set was collected retrospectively from the most recent available TCD scan; this ranged from one to 18 months (median 12 months) before competency, and represented the local TCD practice at each of the three centres. The post-training data set was collected prospectively in the first year following competency and reflected the TCD practice following implementation of a standardised TCD protocol and STOP classification scheme. TCD data were collected from the basal cerebral arteries using imaging and/or non-imaging TCD, depending on local practice. The data included patient demographics, TCD mode and protocols, time-averaged mean of the maximum velocity (TAMMV) from the basal cerebral arteries, STOP classifications, and treatment information.

Continuous variables were described by their means with standard deviations and standard error of the means presented and comparisons made with ANOVA. Categorical variables were presented as relative proportions and comparisons made with Fisher’s exact test.

## 3. Results

### 3.1. Modular Training Programme

Nine training courses were held (six in England, one in Ireland and two in Italy); these were attended by a total of 51 trainees (Table 1). Almost half of the trainees (45%) successfully completed the competency evaluation—the pass rate for clinical scientists with ultrasound experience was 90% (9/10) compared to 46% (13/28) for clinicians with ultrasound experience. Trainees with no previous ultrasound experience, or without access to an ultrasound equipment, did not achieve competency (0/13). Twenty trainees were still eligible to complete their training in the future and eight withdrew from the programme due to problems with local funding for staff or equipment. The ten trainees with an ultrasound background (clinical scientists) were able to acquire TCD skills rapidly, whilst the findings were more variable in the clinician group (surgeons, paediatricians and nurses), with five requiring refresher training courses and twelve failing to complete the minimum annual scan number (forty) due to small local sickle populations.

### 3.2. Comparative Analysis of TCD Data Obtained before and after Training

A total of 441 patients with genotype HbSS were included in this study; 133 patients at Centre 1 (64 males, mean age 8.1 ± 3.7 (range 1.7–15.4 years), 154 patients at Centre 2 (75 males, mean age 7.5 ± 3.2 (range 2–15.1 years), and 154 patients at Centre 3 (74 males, mean age 6.4 ± 3.5 (range 2–15.1 years). There was no significant difference in gender distribution (Chi-Square = 0.015, *p* = 0.99), but more young patients were recruited in Centre 3 (ANOVA, F = 9.7, *p* < 0.001). All those with double heterozygous forms were excluded from the analysis as the TCD programme is only relevant for HbSS.

### 3.3. Pre-Training-STOP Classification

The TCD results at the three centres demonstrated a significant difference in STOP distribution (Fisher’s exact test = 40.0, *p* < 0.001; Table 2). Abnormal STOP classifications were highest in Centre 3, and conditional classifications were higher in Centres 1 and 2. Controlling for treatment regimen did not influence the significance of these findings (Fisher’s Exact treatment = 13.1, *p* < 0.05, and no treatment = 41.6, *p* < 0.001). Non-diagnostic scans, due to either patient movement or attenuation of ultrasound through skull bone, were similar across the three Centres.

Further analysis of the pre-training data set, focusing on the impact of TCD mode on STOP classification, and demonstrated similar STOP distributions between the three Centres for non-imaging TCD studies (Table 3, Fisher’s Exact = 4.8, *p* = 0.26). Only Centres 1 and 3 performed imaging TCD; abnormal and conditional classifications were increased in Centre 3 and abnormal classifications were increased when imaging TCD was used (Table 3; Fisher’s Exact = 16.05, *p* < 0.001).

### 3.4. Pre-Training-Time-Averaged Mean of the Maximum MCA Velocity (TAMMV)

To investigate the disparity in STOP classifications between centres, the source TAMMV velocity data were interrogated. The highest measurable TAMMV from either the right or left MCA in each patient was used in the comparative analysis, as this was the value that was used to determine the STOP classification. Twelve scans were non-diagnostic (Centre 1: 5, Centre 2: 6 and Centre 3: 1). Pre-training, there was a significant difference in the mean TAMMV between Centres; the highest values were obtained at Centre 3 (ANOVA, F = 16.8, *p* < 0.001; Table 4). Imaging TAMMV was significantly higher in Centre 3 (ANOVA, F=17.5, *p* < 0.001) but non-imaging TAMMV showed better agreement between the three Centres (ANOVA, F = 4.2, *p* = 0.02).

### 3.5. Post-Training-STOP Classification

Following training, the STOP distribution across the three centres was more closely aligned (Fischer’s Exact = 6.7, *p* = 0.305; no treatment = 5.6, *p* = 0.41, treatment = 13.8, *p* < 0.001). The variation noted in the treatment group may reflect local criteria for transfusion selection. The same consistency in STOP distribution was demonstrated for non-imaging (Table 3; Fisher’s Exact = 1.3, *p* = 0.99) and imaging TCD data (Fisher’s Exact = 2.6, *p* = 0.42) alone.

### 3.6. Post-Training-Time-Averaged Mean of the Maximum MCA Velocity (TAMMV)

An analysis of the TAMMV data following training confirmed closer agreement of TAMMV obtained from the three Centres for scans performed by either TCD mode (ANOVA F = 1.9, *p* = 0.15), or by imaging TCD alone (ANOVA F = 0.78, *p* = 0.38) or non-imaging TCD alone (ANOVA F = 2.21, *p* = 0.11; Table 4).

## 4. Discussion

Our multi-centre European study demonstrated that one of the barriers to TCD screening for children with SCD in Europe—the lack of skilled personnel—can be overcome through a standardised and reproducible teaching module. The STOP trial provided a comprehensive and reproducible protocol for the systematic assessment of stroke risk in children with SCD using TCD [11]. Successful implementation of the STOP trial results into clinical practice is dependent on adherence to the STOP protocol. In the present study, practitioners recruited onto the training programme were from a range of professional backgrounds, those with previous training in vascular ultrasound demonstrated a rapid achievement of TCD competency.

Practitioners without previous ultrasound experience often required attendance at refresher courses to develop their TCD skills and, as a result, took longer to achieve competency. A common obstacle to achieving competency was the small number of children with SCD in the local hospital population, which limited scanning experience—particularly in trainees who did not perform any other vascular ultrasound assessments [14,15]. The effectiveness and transferable nature of the training programme was demonstrated by its success in producing competent TCD operators from diverse backgrounds in three European countries. A high drop-out was observed, especially among clinicians, who failed to complete the training. This seems to suggest that organising a stable TCD service for children with SCD requires staff that can be dedicated, for a specific amount of time, to TCD evaluation, and that this activity cannot be performed over the top of their clinical commitments if it does not have a dedicated schedule. A specific background does not seem to be necessary in order to be trained but a commitment from the health service and the staff to include a dedicated time for screening of SCD patients is essential.

Before TCD training, the participating haematology clinics in Europe demonstrated a diverse approach to TCD screening. The impact of these protocol variations was responsible for some centres reporting an unusually high number of abnormal STOP classifications—21% compared to the expected incidence of 10%–15% in an unscanned population [9,26]. The reason for this disparity may be two-fold. Firstly, some variation in mean TAMMV at each centre would be expected due to the distribution of stroke risk and the number of patients on transfusion in each patient population. Centres 1 and 2 had TCD screening in place for several years, and are therefore likely to have lower stroke risk populations. The generally higher velocities obtained at Centre 3 may be due to a higher risk population, as TCD screening was only recently introduced. However, it is more likely that this is due to the TCD protocol, specifically the use of Doppler-angle correction, which produces higher velocities than uncorrected values. The training centre (Centre 1) did not observe this disparity as the imaging TCD protocol used was designed to follow the non-imaging STOP method, without Doppler angle correction. Secondly, different STOP velocity thresholds were applied to imaging and non-imaging TCD data, based on previous recommendations, although recent studies have shown that this is not necessary if a standardised TCD technique is used [18,19,20].

This study provided an insight into the variation in TCD practice present in paediatric haematology clinics in Europe. The consistent TAMMV and STOP distribution achieved after the training program when using either imaging or non-imaging TCD, however, indicates that standardisation can be achieved in terms of TCD technique and STOP velocity thresholds. Other investigators have advised caution when using imaging TCD, due to inconsistencies between the techniques [20,21,22,23]. However, due to the ready availability of imaging TCD equipment in many hospitals, it continues to be used for SCD screening [18,19,20,21,22,23]. Our study demonstrates that using imaging TCD without appropriate training may result in under-estimations of stroke risk, due to lower measured velocity or paradoxically over-estimation of stroke risk if Doppler angle-corrected velocities are used or if modified STOP thresholds are applied. Previous studies have shown that the difference between the two methods is not systematic, so that modified thresholds are inappropriate and may result in the erroneous selection or exclusion of children for transfusion therapy. The current study focused on measurements from the middle cerebral artery, where the blood vessel anatomical orientation is most favourable for Doppler investigations. The situation for other basal cerebral arteries, including the terminal internal carotid, anterior and posterior cerebral arteries, is less advantageous, and the discrepancy between techniques may be more pronounced [23]. This study has limitations, in that imaging and non-imaging TCD was not performed in every patient, so that the direct comparison of TAMMV between TCD modes was not possible.

## 5. Conclusions

Diagnostic vascular ultrasound is highly operator-dependent; hence training and competency validation are essential in producing skilled TCD operators. The modular training programme described here was effective in ensuring standardised TCD technique, irrespective of professional background. In this multi-centre study, TCD velocity measurements and STOP classifications were consistent, irrespective of TCD mode and European country. We believe that this is the first modular training programme that has demonstrated efficacy when delivered in different European countries. Delivery of this programme in areas where TCD is underprovided will augment the number of trained TCD operators, thus facilitating access to specialist diagnostic services. This will have a significant impact on public health across Europe, where SCD patients are increasing due to migration. Competency and quality assurance (QA) are important components of such a screening programme; further work is in progress to develop an achievable QA programme for the ongoing regulation of this screening programme.

Among the motivating factors for the present study is the low coverage of TCD globally, and the ongoing requirement for TCD surveillance. The imaging-TCD technique may be the preferred technique to achieve competency and may offer a shorter learning curve, as it aids better vessel localisation and also allows the assessment of the extracranial part of internal carotid. However, with the important caveat that even more meticulous attention to technique is essential to ensure accurate velocity measurement. Our future project is to design and organise ongoing audit of TCD practitioners to ensure consistency and prospective follow up data for patient outcomes in SCD.

## Figures and Tables

**Table 1 jcm-09-00044-t001:** Results of Transcranial Doppler Ultrasound (TCD) competency evaluation of the 51 trainees who attended the modular TCD training programme.

Centre	Trainee Designation	*n*	Ultrasound Experience	TCD Competency			
				Passed (%)		Pending	Withdrawn
1	Clinical Scientist	6	6	6	100%	0	0
	Clinician	15	8	7	47%	6	2
	Nurse	6	0	0	0%	0	6
	Total	27	14	13	(48%)	6	8
2	Clinical Scientist	2	2	2	100%	0	0
3	Clinical Scientist	2	2	1	50%	1	0
	Clinician	20	20	7	35%	13	0
	Total	22	22	8	(36%)	14	0
all		51	38	23	(45%)	20	8

**Table 2 jcm-09-00044-t002:** The STOP distribution at the three centres, prior to operators attending the TCD training course and including both imaging and non-imaging TCD studies.

Centre	Stop Category (Pre-Training)				
	Normal	Conditional	Abnormal	Non-Diagnostic	Total
1	101	20	8	4	133
	(76%)	(15%)	(6%)	(3%)	
2	136	12	0	6	154
	(88%)	(8%)		(4%)	
3	74	16	20	3	113
	(65%)	(14%)	(18%)	(3%)	
all	311	48	28	13	400

**Table 3 jcm-09-00044-t003:** Distribution of STOP categories with respect to Centre and TCD mode, before and after TCD training.

Centre	TCD Mode	Stop Category									
		Before-TCD Training					After-TCD Training				
		Normal	Conditional	Abnormal	Non-Diagnostic	Total	Normal	Conditional	Abnormal	Non-Diagnostic	Total
1	non-imaging	33	2	0	0	35	32	2	0	1	35
		(94%)	(6%)				(91%)	(6%)		(3%)	
	imaging	68	18	8	4	98	82	11	1	4	98
		(70%)	(18%)	(6%)	(4%)		(84%)	(11%)	(1%)	(4%)	
2	non-imaging	136	12	0	6	154	134	8	6	6	154
		(88%)	(8%)		(4%)		(87%)	(5%)	(4%)	(4%)	
3	non-imaging	44	9	0	1	54	38	2	1	1	42
		(81%)	(17%)		(2%)		(83%)	(5%)	(2%)	(2%)	
	imaging	30	7	20	2	59	39	3	1	0	43
		(51%)	(12%)	(34%)	(3%)		(91%)	(7%)	(3%)		
all	non-imaging	213	21	0	7	241	204	12	7	8	231
		(88%)	(9%)		(3%)		(88%)	(5%)	(3%)	(4%)	
	imaging	98	25	28	6	157	121	14	2	4	141
		(62%)	(16%)	(18%)	(6%)		(86%)	(10%)	(1%)	(3%)	

**Table 4 jcm-09-00044-t004:** Time-averaged mean of the maximum velocity (TAMMV) from the middle cerebral artery (MCA) obtained before (PRE) and after (POST) TCD training. Upper panel—all TCD data, centre panel—non-imaging TCD velocity data and lower panel imaging TCD velocity data.

TCD Mode	Centre	PRE-Training				POST-Training			
		N	TAMMV MCA (cm/s)			N	TAMMV MCA (cm/s)		
			Mean	SD	SE		Mean	SD	SE
all	1	128	132	27	2.4	128	132	25	2.2
	2	148	129	24	2.0	148	135	28	2.3
	3	110	155	57	5.4	84	128	32	3.5
non-imaging	1	35	120	22	3.6	47	127	25	4.1
	2	148	129	24	1.8	148	135	28	2.3
	3	53	139	28	3.8	21	127	29	6.3
imaging	1	94	134	29	2.9	94	133	24	2.5
	3	57	169	72	9.5	23	125	29	6

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
