# Peer review of "An Educational Study Promoting the Delivery of Transcranial Doppler Ultrasound Screening in Paediatric Sickle Cell Disease: A European Multi-Centre Perspective"

_jcm, 2019, doi:10.3390/jcm9010044_

Round 1
Reviewer 1 Report
This manuscript reports on an educational project that intended to spread the skills to analyse cerebral blood flow in patients with SCD by transcranial Doppler ultrasound (TCD). Identifying patients at risk of stroke and consecutively preventing stroke is one of the most important issues in the care for patients with SCD. However, TCD is most likely still underused in most countries, rendering such projects highly relevant.
While the study shows some effect of training on patient classification according to the STOP criteria, it does not show an effect on the important clinical endpoints of stroke and regular red blood cell transfusion. Before being considered for publication, the manuscript needs some major revisions:
The focus of the report is on physician training. However, the training course is not described in detail. Please give details on the "modular TCD training programme", including an agenda (which topics were taught for how long? How much time was used for theory and for practice?). Ideally, training material (lecture notes, documentation of the ultrasound examination by video) should be provided online. This material should be suited both for physicians to acquire basic TCD skills and to guide other institutions in establishing such a training programme.
"Almost half of the trainees (45%) successfully completed the competency evaluation". This is a disappointingly low number. At what point did the failing trainees leave the program? Was it already after the two day seminar or was it during the 40 training scans? How and at which point was competency assessed? Should there be an "interim assessment", e.g. after the first 10 training scans?
The authors demonstrate how the percentage of normal, conditional and abnormal change with training. However, the main goal is not only to get reproducible TCD readings, but to prevent stroke by giving regular red blood cell transfusion to those at risk. Please show how many patients with normal, conditional and abnormal TCD suffered from stroke before and after the training and how many received regular transfusion for stroke prevention. These data could be included in Table 2 (in the current version of the manuscript, table 2 only shows data that are also contained in table 3 and should either be enhanced with additional data or deleted).
Table 3: The number of patients with abnormal TCD after training is low (only 2%). What proportion of patients is already under transfusion? Does this proportion change from before training to after training?
The references should be revised and selected with more diligence:
introduction: "less than 50% of children with SCD have access to TCD". However, none of the cited references supports the claim that TCD is used in <50% of children with SCD in Europe, although this may well be true: ref 6 is only about newborn screening for SCD, 14 is only about HbSC disease, 16 shows that in Padua all children have a TCD and 17 does not comment on TCD usage. If the authors cannot cite conclusive literature on the usage of TCD, I would rather phrase: "Data on the proportion of children who are examined by TCD are scarce." Please include Brousse et al. 2019 (doi 10.3390/jcm8101594) who demonstrate that in France, >90% of children with SCD have an annual TCD examination.
Methods: "previous studies that have shown good agreement between
velocity measured by imaging and non-imaging TCD modalities." Ref 10 does not support this claim.
minor points:
Abstract: A p-value for "no treatment" and "treatment" is given. Is treatment red blood cell transfusion? Please clarify.
In General: For each statistic test, not only a p-value, but also another number is given, e.g. in abstract "Fisher’s Exact: no treatment =4.4, p=0.63". What is the 4.4?
Introduction: "this portable educational program"- What do you mean by "portable"?
Methods: Did any of the centers use sedation for TCD in young children?
Results:
"the pass rate for clinical scientists with ultrasound experience was 90%
(9/10) compared to 46% (13/28) for clinicians with ultrasound experience."- rather: "compared to 46% (13/28) for clinicians without ultrasound experience."
I was surprised by the low proportion of male patients at all centers (less than one third). Is there any good explanation or even speculation on this?
"154 patients at Centre 3 (50 males, mean age 6.4±3.5 (range 2-15.1 years), genotypes: 154 HbSS, 10HbSC, 16 HbSβ thalassemia)"- the three genotypes sum up to 180, not 154 patients at center 3.
Generally, TCD screening is not recommended for HbSC disease because of the low risk of stroke. What was the reason to include HbSC patients and what were the results? Did any of the HbSC patients receive prophylactic red blood cell transfusion as a consequence of abnormal TCD? Did any HbSC patient have a stroke?
Pre-training - STOP classification "Controlling for treatment regimen did not influence the significance of these findings" - what is meant by "treatment"?
Pre-training - Time-averaged mean of the maximum MCA velocity (TAMMV) "Imaging TAMMV was significantly higher in Centre 3 (ANOVA, F=24.5, p<0.001)" Was this still true if looking at HbSS patients only?
Post-training – STOP classification "The variation noted in the treatment group may reflect local criteria for transfusion selection." I was startled by this speculation. Wasn't the aim of the training procedure a harmonization of the indication for regular red blood cell transfusion? It is important to show the proportion of transfused patients in each center before and after training. If there are "local criteria for transfusion selection", please mention these criteria in the methods section.
Discussion:
"A specific background does not seem to be necessary in order to be trained". As all nurses failed the training and mainly physicians who already had ultrasound experience passed the training, I cannot follow this conclusion.
Author Response
While the study shows some effect of training on patient classification according to the STOP criteria, it does not show an effect on the important clinical endpoints of stroke and regular red blood cell transfusion. Before being considered for publication, the manuscript needs some major revisions:
The focus of the report is on physician training. However, the training course is not described in detail. Please give details on the "modular TCD training programme", including an agenda (which topics were taught for how long? How much time was used for theory and for practice?). Ideally, training material (lecture notes, documentation of the ultrasound examination by video) should be provided online. This material should be suited both for physicians to acquire basic TCD skills and to guide other institutions in establishing such a training programme. Further detail has been provided about the training programme including the amount of time spent on lectures and practical hands-on training of TCD skills and competency evaluation.
"Almost half of the trainees (45%) successfully completed the competency evaluation". This is a disappointingly low number. At what point did the failing trainees leave the program? Was it already after the two day seminar or was it during the 40 training scans? How and at which point was competency assessed? Should there be an "interim assessment", e.g. after the first 10 training scans? Trainees who failed to complete the training course completed the 2-day training programme but failed to either obtain equipment or scan patients at their hospital.
The authors demonstrate how the percentage of normal, conditional and abnormal change with training. However, the main goal is not only to get reproducible TCD readings, but to prevent stroke by giving regular red blood cell transfusion to those at risk. Please show how many patients with normal, conditional and abnormal TCD suffered from stroke before and after the training and how many received regular transfusion for stroke prevention. There were no strokes in this group of patients. The essence of this paper however is to demonstrate that training programme is effective is skills acquisition and reproducibility of the technique. The question about the outcome of patient treatment is beyond the remit of this paper.
These data could be included in Table 2 (in the current version of the manuscript, table 2 only shows data that are also contained in table 3 and should either be enhanced with additional data or deleted).
Table 3: The number of patients with abnormal TCD after training is low (only 2%). What proportion of patients is already under transfusion? Does this proportion change from before training to after training? 11% of patients (27/243) received transfusion in the pre-training group compared to 14% (24/174) in the post-training group. The interval between the two studies was kept to the barest minimum and all re-scanning were done within a twelve month’s period intended to produce comparable cerebral velocities
The references should be revised and selected with more diligence:
introduction: "less than 50% of children with SCD have access to TCD". However, none of the cited references supports the claim that TCD is used in <50% of children with SCD in Europe, although this may well be true: ref 6 is only about newborn screening for SCD, 14 is only about HbSC disease, 16 shows that in Padua all children have a TCD and 17 does not comment on TCD usage. If the authors cannot cite conclusive literature on the usage of TCD, I would rather phrase: "Data on the proportion of children who are examined by TCD are scarce." Please include Brousse et al. 2019 (doi 10.3390/jcm8101594) who demonstrate that in France, >90% of children with SCD have an annual TCD examination.
Methods: "previous studies that have shown good agreement between
velocity measured by imaging and non-imaging TCD modalities." Ref 10 does not support this claim.
The references have been revised accordingly including Brousse et al 2019
minor points:
Abstract: A p-value for "no treatment" and "treatment" is given. Is treatment red blood cell transfusion? Please clarify.
Treatment was paced red cell transfusion either as simple top up or red cell exchange trasfusion programme which is the treatment for abnormal TCD (more than or equal to 200cm/sec
In General: For each statistic test, not only a p-value, but also another number is given, e.g. in abstract "Fisher’s Exact: no treatment =4.4, p=0.63". What is the 4.4?
Introduction: "this portable educational program"- What do you mean by "portable"?- this has been changed to ‘modular’ which was the intended word referring to the nature of training to ensure a sustainable outcome.
Methods: Did any of the centers use sedation for TCD in young children? NO sedation was allowed for any of the centres. Sedation is not recommended for stroke screening as this would complicate the result interpretation by influencing cerebrovascular flow velocity.
Results:
"the pass rate for clinical scientists with ultrasound experience was 90%
(9/10) compared to 46% (13/28) for clinicians with ultrasound experience."- rather: "compared to 46% (13/28) for clinicians without ultrasound experience."
I was surprised by the low proportion of male patients at all centers (less than one third). Is there any good explanation or even speculation on this?
we have answered this in the text.
"154 patients at Centre 3 (50 males, mean age 6.4±3.5 (range 2-15.1 years), genotypes: 154 HbSS, 10HbSC, 16 HbSβ thalassemia)"- the three genotypes sum up to 180, not 154 patients at center 3.
We have removed all non HbSS/HbSbeta o phenotypes as suggested. The reason this was included in the original because the main objective of the study was to establish a reproducible training programme to validate against the programme by the original training centre used for the STOP trial.
Generally, TCD screening is not recommended for HbSC disease because of the low risk of stroke. What was the reason to include HbSC patients and what were the results? Did any of the HbSC patients receive prophylactic red blood cell transfusion as a consequence of abnormal TCD? Did any HbSC patient have a stroke?
Pre-training - STOP classification "Controlling for treatment regimen did not influence the significance of these findings" - what is meant by "treatment"?
Pre-training - Time-averaged mean of the maximum MCA velocity (TAMMV) "Imaging TAMMV was significantly higher in Centre 3 (ANOVA, F=24.5, p<0.001)" Was this still true if looking at HbSS patients only?-
we reported this in our new submission.
Post-training – STOP classification "The variation noted in the treatment group may reflect local criteria for transfusion selection." I was startled by this speculation. Wasn't the aim of the training procedure a harmonization of the indication for regular red blood cell transfusion?
This statement is meant to convey the fact that TCD screening was introduced to the Centre 3 for the first time in contrast to centre where TCD screening has been regular standard care over 15 years. It is therefore expected that the ratio of those abnormal or conditional TCD would reduce as these are repeated screening of normal velocities. It is really unsurprising that over time those with abnormal TCD would fall.
It is important to show the proportion of transfused patients in each center before and after training. If there are "local criteria for transfusion selection", please mention these criteria in the methods section. The criteria for selection for blood transfusion is a global standard based on the Trial reported by Adam et al, Stroke prevention trial in sickle cell anaemia. Control Clin Trials. 1998 Feb; 19(1):110-29.PMID: 9492971 and N Engl J Med. 1998 Jul 2; 339(1):5-11. PMID: 9647873. Those with velocities 200cm/sec based on the non-imaging techniques are offered regular blood transfusion in order to maintain Haemoglobin S percentage of less than 30%.
Discussion:
"A specific background does not seem to be necessary in order to be trained". As all nurses failed the training and mainly physicians who already had ultrasound experience passed the training, I cannot follow this conclusion.
This was not the intended comment and has now been deleted.

Reviewer 2 Report
The article by Inusa et al presents a multi-center study designed to promote delivery of standardized TCD screening in SCD patients.
The starting point of the study is interesting (50% of patients do not benefit from appropriate screening) but unfortunately the conclusions of the study do not contribute much.
2 major methodological problems are to be regretted:
1-Inclusion of patients other than SS. Indeed, SC patients do not have the same risk of cerebral vasculopathy at all, and I do not understand why they were included in the study. Since their proportion is different between the different centres (less important in centre 3, which most probably influences the over-representation of pathological examinations), this totally biases the results, without this point ever being discussed by the authors. The same applies to Sbeta-thalassemic patients, whose authors do not even specify whether they are Sbeta0-thalassemic patients (at risk of cerebral vasculopathy) or or Sbeta+ patients (very low risk of cerebral vasculopathy). I think only SS patients should have been included in the study.
2- the authors specify that some trainees did not complete the training cycle at the time of writing the article. Their proportion is quite high (up to 2/3 in some groups), and I don't understand why the authors didn't wait until all trainees had completed the training to analyze the results.
Finally, statistical analyses should appear in the tables because it is currently difficult to understand which results are significant and which are not.
In total, the study concludes that trainees without previous TCD experience took longer to achieve competency, and that people need to be trained so that screening can be done better. Unfortunately, I am not sure that these conclusions are not already known.
Author Response
we have now addressed all the points raised in the revised manuscript including a re-analysis of the data to remove non -HbSS/ SBetao.
we have also explained some of the key factors we though influenced the outcome for trainees.
This has improved our paper tremendously, thanks.
Round 2
Reviewer 1 Report
The authors have adressed few of my comments from round 1. In addition, it is difficult to follow the changes because the point-by-point response does not cite what has been changed in the text. I restrict myself to the comments I had made before:
The focus of the report is on physician training. However, the training course is not described in detail. Please give details on the "modular TCD training programme", including an agenda (which topics were taught for how long? How much time was used for theory and for practice?). Ideally, training material (lecture notes, documentation of the ultrasound examination by video) should be provided online. This material should be suited both for physicians to acquire basic TCD skills and to guide other institutions in establishing such a training programme.
Further detail has been provided about the training programme including the amount of time spent on lectures and practical hands-on training of TCD skills and competency evaluation.
The few lanes that have been inserted will neither help physicians acquire basic TCD skills nor guide other institutions in establishing such a training programme.
"Almost half of the trainees (45%) successfully completed the competency evaluation". This is a disappointingly low number. At what point did the failing trainees leave the program? Was it already after the two day seminar or was it during the 40 training scans? How and at which point was competency assessed? Should there be an "interim assessment", e.g. after the first 10 training scans?
Trainees who failed to complete the training course completed the 2-day training programme but failed to either obtain equipment or scan patients at their hospital.
OK
The authors demonstrate how the percentage of normal, conditional and abnormal change with training. However, the main goal is not only to get reproducible TCD readings, but to prevent stroke by giving regular red blood cell transfusion to those at risk. Please show how many patients with normal, conditional and abnormal TCD suffered from stroke before and after the training and how many received regular transfusion for stroke prevention. These data could be included in Table 2 (in the current version of the manuscript, table 2 only shows data that are also contained in table 3 and should either be enhanced with additional data or deleted).
There were no strokes in this group of patients. The essence of this paper however is to demonstrate that training programme is effective is skills acquisition and reproducibility of the technique. The question about the outcome of patient treatment is beyond the remit of this paper.
Please mention in the text that there were no strokes.
Still, Table 2 can be deleted because the data are shown in more detail in table 3.
Table 3: The number of patients with abnormal TCD after training is low (only 2%). What proportion of patients is already under transfusion? Does this proportion change from before training to after training?
11% of patients (27/243) received transfusion in the pre-training group compared to 14% (24/174) in the post-training group.
Please mention these numbers in the text.
The interval between the two studies was kept to the barest minimum and all re-scanning were done within a twelve month’s period intended to produce comparable cerebral velocities
Please mention in the text that the same patients were analysed before and after training with the shortest reasonable interval.
The references should be revised and selected with more diligence:
introduction: "less than 50% of children with SCD have access to TCD". However, none of the cited references supports the claim that TCD is used in <50% of children with SCD in Europe, although this may well be true: ref 6 is only about newborn screening for SCD, 14 is only about HbSC disease, 16 shows that in Padua all children have a TCD and 17 does not comment on TCD usage. If the authors cannot cite conclusive literature on the usage of TCD, I would rather phrase: "Data on the proportion of children who are examined by TCD are scarce." Please include Brousse et al. 2019 (doi 10.3390/jcm8101594) who demonstrate that in France, >90% of children with SCD have an annual TCD examination.
Methods: "previous studies that have shown good agreement between velocity measured by imaging and non-imaging TCD modalities." Ref 10 does not support this claim.
The references have been revised accordingly including Brousse et al 2019
OK
minor points:
Abstract: A p-value for "no treatment" and "treatment" is given. Is treatment red blood cell transfusion? Please clarify.
Treatment was paced red cell transfusion either as simple top up or red cell exchange trasfusion programme which is the treatment for abnormal TCD (more than or equal to 200cm/sec
Please explain in the text. The simplest way would be “on chronic transfusion” vs. “no chronic transfusion”
In General: For each statistic test, not only a p-value, but also another number is given, e.g. in abstract "Fisher’s Exact: no treatment =4.4, p=0.63". What is the 4.4?
?
Introduction: "this portable educational program"- What do you mean by "portable"?-
this has been changed to ‘modular’ which was the intended word referring to the nature of training to ensure a sustainable outcome.
OK
Methods: Did any of the centers use sedation for TCD in young children?
NO sedation was allowed for any of the centres. Sedation is not recommended for stroke screening as this would complicate the result interpretation by influencing cerebrovascular flow velocity.
Please mention in the text. If centers had used sedation before but not after training, this would have been an important confounder.
Results:
"the pass rate for clinical scientists with ultrasound experience was 90% (9/10) compared to 46% (13/28) for clinicians with ultrasound experience."- rather: "compared to 46% (13/28) for clinicians without ultrasound experience."
Please correct!
I was surprised by the low proportion of male patients at all centers (less than one third). Is there any good explanation or even speculation on this?
we have answered this in the text.
The authors now focused on HbSS patients and suddenly now the gender distribution is even. I do not understand why compound heterozygous patients should have been predominantly female- this makes me really worry about the data quality.
"154 patients at Centre 3 (50 males, mean age 6.4±3.5 (range 2-15.1 years), genotypes: 154 HbSS, 10HbSC, 16 HbSβ thalassemia)"- the three genotypes sum up to 180, not 154 patients at center 3.
Generally, TCD screening is not recommended for HbSC disease because of the low risk of stroke. What was the reason to include HbSC patients and what were the results? Did any of the HbSC patients receive prophylactic red blood cell transfusion as a consequence of abnormal TCD? Did any HbSC patient have a stroke?
We have removed all non HbSS/HbSbeta o phenotypes as suggested. The reason this was included in the original because the main objective of the study was to establish a reproducible training programme to validate against the programme by the original training centre used for the STOP trial.
Still, the numbers are not consistent. In the text, the authors mention 154 patients at center 3, in table 2 and 3, only 113 patients at center 3 are shown.
Pre-training - STOP classification "Controlling for treatment regimen did not influence the significance of these findings" - what is meant by "treatment"?
?
Pre-training - Time-averaged mean of the maximum MCA velocity (TAMMV) "Imaging TAMMV was significantly higher in Centre 3 (ANOVA, F=24.5, p<0.001)" Was this still true if looking at HbSS patients only?-
we reported this in our new submission.
OK
Post-training – STOP classification "The variation noted in the treatment group may reflect local criteria for transfusion selection." I was startled by this speculation. Wasn't the aim of the training procedure a harmonization of the indication for regular red blood cell transfusion?
This statement is meant to convey the fact that TCD screening was introduced to the Centre 3 for the first time in contrast to centre where TCD screening has been regular standard care over 15 years. It is therefore expected that the ratio of those abnormal or conditional TCD would reduce as these are repeated screening of normal velocities. It is really unsurprising that over time those with abnormal TCD would fall.
Explain in the text .
It is important to show the proportion of transfused patients in each center before and after training. If there are "local criteria for transfusion selection", please mention these criteria in the methods section.
The criteria for selection for blood transfusion is a global standard based on the Trial reported by Adam et al, Stroke prevention trial in sickle cell anaemia. Control Clin Trials. 1998 Feb; 19(1):110-29.PMID: 9492971 and N Engl J Med. 1998 Jul 2; 339(1):5-11. PMID: 9647873. Those with velocities 200cm/sec based on the non-imaging techniques are offered regular blood transfusion in order to maintain Haemoglobin S percentage of less than 30%.
If criteria for transfusion were universal, do not write about “local criteria”.
Discussion:
"A specific background does not seem to be necessary in order to be trained". As all nurses failed the training and mainly physicians who already had ultrasound experience passed the training, I cannot follow this conclusion.
This was not the intended comment and has now been deleted.
OK
Reviewer 2 Report
The study has become clearer since the exclusion of SC patients.
Nevertheless, I am not convinced by the interest of the study, whose only conclusion in the end is to demonstrate that more staff must be trained in TCD so that it can be done better.
Author Response
We have probably failed to highlight the relevance of our study in the conclusions. We have change the conclusions in the abstract:
This TCD training programme for SCD was shown a standardised and reproducible teaching module which can be easily implemented in different European countries. Furthermore it allowed to achieve standardization of TCD technique and STOP velocity thresholds in order to improve the access to advanced treatment for children with SCD.
This manuscript is a resubmission of an earlier submission. The following is a list of the peer review reports and author responses from that submission.
Round 1
Reviewer 1 Report
Interesting article with a clear objective: to improve the availability of TCD adapted to children with sickle cell disease to decrease the incidence of stroke in this population. Current difficulties are well identified: two types of equipment (blind and imaging TCD), no consensus about velocities thresholds… The design is adequate with the comparison between different centres with long and short experience, that allows detecting discrepancies in practice in non sufficiently trained beginners. Improvement of the conclusion is needed with recommendations about the technical aspects and standardization of TCD in SCA children.
A few comments
L 62 SCD or SCA . see comment L 174 below
L 131 Please specify that no angle correction was applied in the post-training protocol.
L 174 TCD scanning of HbSC patients may be usefull for trainees to acquire experience but TCD screening in recommended only in Sickle Cell Anemia patients, i.e Hb SS and Hb SB0 genotypes in US and France. Please add a comment in the discussion paragraph
dischttps://www.nhlbi.nih.gov/health-topics/evidence-based-management-sickle-cell-disease; https://www.has-sante.fr/plugins/ModuleXitiKLEE/types/FileDocument/doXiti.jsp?id=c_938888
L 200 Could you please specify in the methods paragraph which arteries have been selected for classification . Only MCAs? Or also terminal ICAs and ACAs ? In the conclusion of the article, it seems relevant to propose inclusion of ACA and terminal ICA as well as the extracranial part of the internal carotid artery in classification when updating this protocol.
L 269 The conclusion could be more detailed. This is an opportunity to focus on the main technical tips for beginners (TAMV, no angle correction, same cut-off for imaging and non imaging TCD , screened arteries…). It could be interesting to repeat that the learning curve is shorter for practitioners with previous training in vascular ultrasound, who should be targeted first for training.
Reviewer 2 Report
Performance of regular evidence-based care for patients with sickle cell disease such as annual transcranial doppler screening is an important clinical aspect of stroke prevention and more generally quality assurance. The goal of this report was to describe the outcomes of a 3-site QA in TCDs following training, assuming that a paucity of trained staff accounted for some of the incomplete surveillance. Overall there were interesting results over a large number of patients and multiple sites. However, there is considerable need for methodological details. Also, data were not shown regarding the proportion of those with prior training, and how their outcomes differed vs. those without prior training.
Introduction:
The statement that SCD is the most common “genetic disease” in France and the UK should clarify among single gene disorders. Sentence lines 69-70: “There is no formal 69 training centre outside of the USA, which contributes to…” should use the term “that” rather than “which,” as it refers to a formal training center rather than the USA.
Methods:
More information is needed about who were the trainees and how were they selected. Pediatric providers? Specialists? Did they volunteer? Etc. What were the differences between the 3 centers? How much training and experience had they previously received? There was only a terse statement in the Discussion about who they were. How was quality assurance assessed among the 40 training scans per trainee? Were their specific subjects upon which the trainees could practice? Was there direct observation, video or other confirmation of technique? As the authors likely know, missing an aberrancy can be easier than finding it. Who were the patients being scanned? Volunteers? A random sample of patients? Other? Were the same patients scanned by >1 trainee? Why were some have sickle hemoglobinopathies beyond HbSS? What normal values have been established for them? For analyses, the authors should have examined the correlation (kappa) between pre- and post- examination and trainees within sites. And comparison data should, where possible, be placed within the table to which it refers.
Results:
There was a high level of trainee drop-out, ~20%. Nearly all surgeons. How randomly did drop-outs occur? Why were so many patients found to be in the pre-testing abnormal range, especially at site 3? 17% seems very high. Does that finding reflect some issue among the surgeons? Table 2 vs. 3: did the same patients have conditional TAMMV in the pre- vs. post-training set? Did TCD pre- and post-training accuracy track within each trainee, e.g. those with prior experience and training? Data not shown.
a
Conclusions:
Are there certain staff profiles to suggest who should be trained? For example, should surgeons be omitted, given high drop out? Or might they need workload accommodations or other? Do these results have implications for the burgeoning approach to TCD in Sub-Saharan Africa?
Minor points:
Abstract:
The data in parentheses stating “treatment” rather than “training” is confusing. The STOP study acronym will likely not be widely understood without explanation. The topic in the conclusion about an “imaging background” should be introduced earlier in the Abstract.
Introduction:
The text would be easier to read if broken into several paragraphs. Several statements are made without references, especially regarding the reasons for poor TCD screening rates.
Tables:
Tables 2 and 3 should have the same format. Table 3 is not formatted to be conducive for rapid understanding of the data. In comparison, tables 1 and 2 are readily readable.